# Does Overhead Squat Performance Affect the Swing Kinematics and Lumbar Spine Loads during the Golf Downswing?

**DOI:** 10.3390/s24041252

**Published:** 2024-02-15

**Authors:** Zi-Han Chen, Marcus Pandy, Tsung-Yu Huang, Wen-Tzu Tang

**Affiliations:** 1MSc and MPE Dual Programme in International Sport Coaching Science, National Taiwan Sport University, Taoyuan City 33301, Taiwan; jasonzhchen@gmail.com; 2MSc and MPE Dual Programme in International Sport Coaching Science, University of Physical Education, 1123 Budapest, Hungary; 3Department of Mechanical Engineering, University of Melbourne, Parkville, VIC 3010, Australia; pandym@unimelb.edu.au; 4Graduate Institute of Athletic and Coaching Science, National Taiwan Sport University, Taoyuan City 33301, Taiwan; a3017927@gmail.com

**Keywords:** golf swing, shear force, compressive force, squat, LBP

## Abstract

The performance of the overhead squat may affect the golf swing mechanics associated with golf-related low back pain. This study investigates the difference in lumbar kinematics and joint loads during the golf downswing between golfers with different overhead squat abilities. Based on the performance of the overhead squat test, 21 golfers aged 18 to 30 years were divided into the highest-scoring group (HS, N = 10, 1.61 ± 0.05 cm, and 68.06 ± 13.67 kg) and lowest-scoring group (LS, N = 11, 1.68 ± 0.10 cm, and 75.00 ± 14.37 kg). For data collection, a motion analysis system, two force plates, and TrackMan were used. OpenSim 4.3 software was used to simulate the joint loads for each lumbar joint. An independent *t*-test was used for statistical analysis. Compared to golfers demonstrating limitations in the overhead squat test, golfers with better performance in the overhead squat test demonstrated significantly greater angular extension displacement on the sagittal plane, smaller lumbar extension angular velocity, and smaller L4-S1 joint shear force. Consequently, the overhead squat test is a useful index to reflect lumbar kinematics and joint loading patterns during the downswing and provides a good training guide reference for reducing the risk of a golf-related lower back injury.

## 1. Introduction

Golf has become an increasingly popular sport worldwide and is viewed as an enjoyable and practical means of staying active throughout an individual’s lifetime. Previous studies have demonstrated that playing golf provides an adequate amount of physical activity to improve overall health and well-being, especially for elderly golfers whose physiological training threshold is lowered by age [1,2]. While golf can provide some benefits for general health and fitness, the sport also appears to have particular risks of injury that may significantly affect players’ enjoyment of the activity [3].

Along with its popularity, the injury rate for golfers has indeed increased steadily over the years, with golf-related low back pain (LBP) being the most common injury [4,5,6]. Research has explored the relationships between spine conditions and golf performance. One-third of recreational players reported that LBP has a negative effect on golf games [2]. Additionally, the prevalence of lower back injuries has been estimated to be between 15% and 35% in amateurs and up to 55% in professional golf players [7]. There are several factors that may contribute to LBP, including poor swing mechanics and fatigue due to overuse [8]. Rapidly swinging a club is a crucial part of the golf game. In order to create a potential advantage at the beginning of the competition, golfers tend to dedicate considerable practice time for swings each day to generate a fast and powerful swing. Poor swing mechanics, combined with overuse, may ultimately increase the risk of LBP for golfers. Furthermore, along with improper swing mechanics, suboptimal physical fitness can generate considerable or abnormal forces localized in the lumbar region [9]. This can cause significant muscle spasms due to back muscle strain or spinal ligament sprain, which usually leads to the development of LBP.

To identify physical deficiencies that are critical to golf swing and injury prevention, the world’s leading golf education organization, the Titleist Performance Institute (TPI), has developed a golf-specific physical screening system similar to the Functional Movement Screen (FMS^TM^, Chatham, VA, USA). The TPI has identified some inappropriate swing mechanics that they categorize as “swing faults” to help coaches and golfers better understand swing mechanics and improve their game.

The FMS^TM^ is composed of seven fundamental movement patterns (tests) that require a balance of mobility and stability [10]. The overhead squat (OHS) is one such test that the FMS^TM^ uses to assess bilateral, symmetrical, and functional mobility of the hips, knees, ankles, shoulders, and thoracic spine, as well as the stability and motor control of core musculature [11]. The TPI also uses the OHS test as one of its movement screens to assess golfers’ strength, flexibility, and balance [12]. An individual with restrictions on spine mobility, hip mobility, or core motor function may fail the OHS test.

Studies on OHS performance have identified some swing faults that are documented by the TPI [12]. One of the most common swing faults among amateur golfers is known as “loss of posture” [13], where the golfer has changed the knee flexion angle, the trunk flexion angle, or the head position between their address posture and impact position [12]. Another common swing fault is “slide,” which is an excessive lateral shift of the hips toward the target on the downswing. Gulgin and his colleagues found that golfers with low overhead squat ability were two to three times more likely to exhibit early hip extension, loss of posture, or slide during the swing in comparison to golfers who could correctly perform an OHS [12]. They further suggested that common swing faults are linked to inconsistent ball striking and reduced performance [12]. Speariett and Armstrong found that the overhead squat is one of the most difficult tests for amateur golfers to perform, so much so that participants who were unable to perform the overhead squat most commonly presented with a loss of posture (90%) and slide (80%) [13].

The mechanics of the spine during a golf swing in golfers with or without LBP have been well established. Compared to healthy golfers, those with LBP may generate more lateral bending accompanied by the flexion of the spine during the downswing phase [14,15]. Fortunately, professional golfers possess the capability to minimize the recurrence of injuries through technical adjustments [16,17]. Grimshaw and Burden reported the successful elimination of golf-related LBP in professional golfers, partly by reducing the amount of trunk flexion and by adopting a side bend during the downswing [16]. The side bend with trunk flexion can limit the amount of trunk rotation available during the golf swing and may apply more shear force to the spine, thus increasing the risk of injury [18]. The physical requirements of the golf swing may be similar to that of the OHS. Likewise, limitations in the mobility of the hip and spine or weakness in the core muscles may lead to the loss of posture and slide among golfers.

Golfers with LBP have shown less hip and spine mobility and delayed core muscle activation compared to healthy golfers [1,19,20]. Considering the fact that both the golf swing and OHS demand a normal function of the core, lower limb, and shoulder mobility in the three-dimensional space, the FMS overhead squat, which also involves core, lower limb, and shoulder mobility, is likely a useful test that can assess all of these elements simultaneously [11]. It may therefore be possible to prevent LBP by using the OHS as a test to assess players’ spinal biomechanics during the golf downswing. To the best of our knowledge, no study to date has investigated the impact of overhead squat abilities on the biomechanical variables of lumbar spine flexion and lateral bending during the golf swing.

To understand the mechanisms of LBP, scientists have developed different methodologies that measure the lumbar joint loads during the golf swing. For example, Hosea and his colleagues made the first attempt to estimate lumbar spinal loads during a golf swing using four video cameras at 30Hz to modeling the lumbar spine loads by Arial Performance Analysis System, and they found that the lumbar spine shear force during the golf swing was 80 percent greater in amateurs than in professionals, whereas the compressive force for both groups was more than eight times body weight (BW) [21]. However, the model’s simplicity and the lack of ground reaction forces for the calculation of kinetic parameters are limitations that may inhibit the validity of the obtained lumbar load.

In another study, Lim and his colleagues used four super VHS camcorders at 60 Hz and two force plates to analyze the three-dimensional (3D) kinematics and kinetics of golf swings, and the average EMG levels for different phases of the golf swing were used in the optimization model in order to compute the contact forces acting on the L4-L5 [22]. They found the shear and compression forces steadily increased during the downswing phase, and the shear loads were about 1.6 BW to 0.6 BW, while the peak compressive loads were greater than the 8 BW found in Hosea’s study [21,22]. However, the findings of this previous study should be interpreted with caution because of its small sample size (five subjects). Another limitation of the previous study is that they did not present the joint loads of L5-S1. Considering the fact that the L4-L5 and L5-S1 discs are subject to large compressive loads due to the weight of the trunk and the muscle activity generated during the golf swing [22], the joint loads applied on L5-S1 should be investigated as well.

Although both studies found the lumbar joint loads to increase continuously during the downswing phase, there was less agreement in the lumbar shear loads. To address these conflicting results, the current study used musculoskeletal modeling with an electromyography sensor combined with an optimization approach to determine lumbar joint forces during the downswing.

The primary purpose of this study was to determine differences in lumbar spine kinematics and joint loads during the downswing between golfers who execute a proper OHS and those who do not. A secondary aim was to investigate whether the ability of a golfer to perform the OHS test is related to their golf swing performance. We hypothesized that golfers who could perform the overhead squat properly would have smaller lumbar spine joint loads, joint angular displacements, and joint angular velocities on L1-L2, L2-L3, L3-L4, L4-L5, and L5-S1 joints during the downswing and thus better performance compared to golfers who could not complete the OHS test.

## 2. Materials and Methods

### 2.1. Participants

A total of 21 highly skilled right-handed golfers aged 18 to 30 years volunteered to participate in this study (9 males, 12 females, handicap 2.4 ± 1.5). All participants were free of any musculoskeletal injuries or disease that would have prevented them from performing their normal golf swing motion or impeded their ability to participate in the overhead deep squat screen. The study was conducted with ethics approval from the Human Research Ethics Committee of the local institution, and the participants provided their written informed consent prior to the commencement of testing.

### 2.2. Experimental Protocol

All experiments were conducted in the Biomechanics Laboratory at National Taiwan Sports University. On arrival, each participant was informed of the study’s purpose and the experimental protocol. Testing was divided into two parts. Each golfer’s overhead squat performance was first measured using an FMS^TM^ kit, after which a biomechanical evaluation of the golf swing was performed using three-dimensional video motion capture techniques.

### 2.3. Overhead Squat Test

The verbal test instruction of the overhead squat test was based on the description given by Cook [12], according to which one certified FMS^TM^ instructor executes the test. For the test, as different footwear conditions are unrelated to FMS performance [23], every individual wore their personal sneakers and positioned themselves by placing their feet about shoulder-width apart and in alignment with the sagittal plane. After that, they held onto a rod while keeping their elbows flexed at a 90-degree angle with the rod positioned above their heads. Then, the rod was lifted above by raising both shoulders and straightening the arms. Next, the participants were given directions to crouch down as much as they could, ensuring that their heels remained in touch with the ground, and the dowel stayed directly above them. Each participant was allowed up to three trials to perform the test successfully. Scoring criteria for the OHS (see Table 1) were used to divide all the participants into a high-scoring group (HS: 3 points, Figure 1a) and a low-scoring group (LS: 2 points or 1 point, Figure 1b) for further analysis.

### 2.4. Kinematic and Kinetic Data Collection

The kinematic data of each participant’s golf swing using a driver were recorded using an 11 Eagle Digital high-speed camera system (Motion Analysis Corporation, Santa Rosa, CA, USA) that sampled at 250 Hz. On both sides, anatomical landmarks such as the front of the head, the rear head, cervical 7, thoracic 10, acromion, upper arm, lateral elbow, radius, ulna, third metacarpophalangeal joint, anterior–superior iliac spine, posterior–superior iliac spine, thigh, knee, shank, ankle, medial ankle, toe, and heel were marked with forty-nine retro-reflective markers measuring 10–12 mm in diameter (see Figure 2). During the swing, ground reaction force data from the lead and trailing legs were collected using two force plates (AMTI, Advanced Management Technology Inc., Arlington, VA, USA), with a sampling rate of 1000 Hz. The force information is synchronized with the motion analysis system. A fourth-order low-pass Butterworth filter with a cut-off frequency of 14 Hz was used to filter the kinematic and ground force data. The kinematic and kinetic data were used as input in a musculoskeletal modeling pipeline available in OpenSim to calculate lumbar spine kinematics and joint loading [24].

For the lower lumbar region of the longissimus thoracic, a pair of wireless EMG sensors (Trigno, Delsys Inc., Natick, MA, USA) were placed on the interspace between L1 and L2 on both sides. The EMG and kinematic data were synchronized using a video camera connected to the EMG system, which recorded the golf swing and displayed the images in real time, along with the EMG signals in Delsys-16 EMGworks Software (Delsys Inc., Natick, MA, USA). Electromyographic data were filtered (six-pole Butterworth and bandpass-filtered 25–500 Hz) and full-wave rectified using signal processing software (EMGworks Analysis Software 4.7.9).

Each participant was given 5 min to warm up prior to data collection. The participant was then instructed to stand on the force plates and perform a maximal swing using the driver. Data were collected for 5 trials. The TrackMan (TrackMan IIIe, Vedbaek, Denmark) Doppler radar system was placed behind the ball striking area to measure the speed of the ball after impact.

### 2.5. Computer Simulation

The full-body lumbar spine model (FBLS, https://simtk.org/home/fullbodylumbar, accessed on 30 Dencember 2020) comprising 21 segments, 30 degrees of freedom, and 324 musculotendon actuators was used to simulate each golf swing [25]. Before the motion capture, the marker setting of the FBLS model was modified for scale, so the model and motion capture data could be matched to fit. All data were converted to a usable format, and the generic musculoskeletal model was scaled to match each participant’s body anthropometry [24]. The inverse kinematic routine in OpenSim was then used to minimize the differences between the positions of skin markers on the participants and the virtual markers on the model. This procedure was undertaken in order to achieve a dynamically consistent set of kinematics and kinetics that best matched the experimentally collected data [24].

To investigate the primary aim of this study, the results of inverse kinematics were used to derive the lumbar joint angle during impact in the sagittal and frontal planes, the peak angular velocity, and angular displacement during the downswing phase in the sagittal and frontal planes.

Next, static optimization (SO) was performed to resolve the net joint moments into individual muscle forces at each instant in time. Finally, the joint reaction analysis tool was used to calculate the internal vertebral joint loads [26]. Lumbar spinal loading was calculated by solving the dynamical equations of motion with the input of muscle forces, gravity, and inertia. Moreover, to attenuate the noise contained within the raw marker data, a filtering process was applied during static optimization, using a low-pass sixth-order Butterworth digital filter at a cut-off frequency of 14 Hz, which was determined based on residual analysis [26]. All the loads reported for a given vertebra were those acting upon it from the inferior vertebra. For example, the L5-S1 loads reported are those from S1 acting on L5. The force was calculated using the Newton’s 2nd law:$${\vec{R}}_{L5\text{-}S1\;}=[M{]}_{L5\;}{\vec{a}}_{L5}\;-\;({\vec{R}}_{L4\;}+\sum {\vec{F}}_{\mathrm{muscles}\;}+{\vec{F}}_{\mathrm{gravity}\;})$$
where ${\vec{R}}_{L5\text{-}S1\;}$ is the force applied by the S1 vertebra to the L5 vertebra, $[M{]}_{L5\;}$ is the matrix of inertial properties of the L5 vertebra, ${\vec{a}}_{L5\;}$ is a vector of angular and linear accelerations of the L5 vertebra, ${\vec{R}}_{L4\;}$ is the force applied by the L4 vertebra to the L5 vertebra, and ${\vec{F}}_{\mathrm{muscles}\;}$ and ${\vec{F}}_{\mathrm{gravity}\;}$ are muscle forces and gravitational forces acting on the L5 vertebra. The L5-S1 compressive force was calculated as the component of ${\vec{R}}_{L5\text{-}S1\;}$ parallel to the longitudinal axis of the L5 vertebra with musculoskeletal modeling using OpenSim software [27], which was used for all subsequent analyses. The L5-S1 shear force was calculated the same way but parallel to the anteroposterior axis of the L5 [26]. The peak shear and compressive forces acting at each lumbar spine joint, specifically L1-L2, L2-L3, L3-L4, L4-L5, and L5-S1, were calculated and used in the statistical analyses described below.

Finally, model simulations were validated by comparing the muscle activations calculated in the model against the EMG data measured during the golf swing. The EMG data were normalized by the peak activation measured during the swing phase and were compared to the simulated muscle activations, which were defined between 0 and 1. We compared the average activation of the longissimus thoracic muscle of 4 subjects to the corresponding EMG (Figure 3).

### 2.6. Statistical Analyses

Descriptive statistics were analyzed to assess means and standard deviations between the low-scoring (LS) and high-scoring (HS) groups. A Pearson’s chi-square test was used to compare gender distributions, and an independent *t*-test was used to determine the significant differences in demographic and performance data between different groups. An independent *t*-test was used to examine the differences in all lumbar kinematics and joint loads during the downswing phase of the golf swing between the LS and HS groups. The Pearson correlation was used to compare the measured EMG data and the simulated activation levels through a time series for each muscle to validate the model. Statistical significance was set at *p* < 0.05, and SPSS 20.0 (SPSS, Chicago, IL, USA) statistical software was used for all data analysis. The effect size for normal data were calculated using Cohen’s *d*.

## 3. Results

Pearson’s chi-square test showed no significant difference in the gender distribution of the groups, χ^2^(1, N = 21) = 2.38, *p* = 0.123. No significant difference was found between the LS and HS groups in body weight, height, played years, and best scores, except for the ball speed of the LS group (M = 142.44, SD = 16.95 mph), which was faster than that of the HS group (M = 126.29, SD = 11.52 mph), *p* = 0.02 (Table 2).

For simulation validation, both sides of the longissimus thoracic muscle activations were consistent with the measured EMG data (Figure 3). The correlation of normalized measured EMG data and the simulated activations for the four subjects was 0.72 ± 0.09 on the right lumbar muscle and 0.74 ± 0.21 on the left lumbar muscle.

### 3.1. Lumbar Joint Kinematics during the Downswing

There was no significant difference between the two groups in the sagittal-plane lumbar flexion angle at impact and the sagittal-plane peak flexion angular velocity during the downswing (Table 3). However, the HS group had significantly greater lumbar angular displacement from top to impact (M = 24.36, SD = 7.11°, *p* = 0.03, *d* =1.01) and a smaller peak extension angular velocity during the downswing (M = 40.51, SD = 26.79 °/s) than the LS group (M = 17.72, SD = 5.94°; M = 119.52, SD = 59.23 °/s), *p* < 0.001, *d* = 1.71 (Table 3). In the frontal plane, there was no significant difference between the two groups in lumbar right-side bending angle at impact, angular displacement from top to impact, and peak right-side bending angular velocity during the downswing (Table 4).

### 3.2. Lumbar Joint Loads during the Downswing

Not all lumbar joints were significantly affected by the overhead squat ability in relation to the shear forces applied in the anterior–posterior direction (Table 5). The HS group had significantly lower shear forces applied at L4-L5 (M = 299.54, SD = 37.30 N) (*p* = 0.01, *d* = 1.28) and L5-S1 (M = 407.90, SD = 59.06 N) (*p* = 0.002, *d* = 1.58) during the downswing compared to the LS group (M = 387.19, SD = 89.16 N; M = 525.19, SD = 86.69 N). There was no significant difference between the two groups in the compressive forces applied at L1-L2, L2-L3, L3-L4, L4-L5, and L5-S1 during the downswing (Table 6).

## 4. Discussion

### 4.1. Influence of the OHS on Golf Performance

The primary objective of this study was to investigate whether limitations in performing an overhead squat affect golf swing performance. The participants were divided into two groups depending on their ability to perform the overhead squat. There was no significant difference in the best golfing scores between the two groups; however, the ball speed of the LS group (M = 142.44, SD = 16.95 mph) was significantly greater than that of the HS group (M = 126.29, SD = 11.52 mph). Although restricting the hip and shoulder joints has been shown to impair golf swing performance [28], this study failed to identify a significant difference in performance between elite golfers with different overhead squat abilities. However, our findings did show a difference in swing kinematics and kinetics on the lumbar spine between the two groups. Our results demonstrate that elite golfers with limited squat ability adopt different swing patterns to keep competitive.

Overall, golfing handicap performance is related to driving distance, driving accuracy, approach accuracy, and putting, combined with physical attributes [29]. Furthermore, more than one swing is needed during a competition. Although the LS group had higher ball speed, no significant difference was observed in their overall performance compared to the HS group. This implies that LS golfers with limited hip and spine mobility are more likely to be affected in terms of accuracy and fatigue during golf swings in the later phases of the competition, ultimately impacting their overall score. The OHS test may therefore be an indicator of the efficiency of golf games.

### 4.2. The Influence of the OHS on Lumbar Spine Kinematics in the Sagittal Plane

Contrary to our hypothesis, we found that greater overhead squat ability may result in an advantage for utilizing more lumbar spine angular displacement in the sagittal plane (extension) to perform the golf downswing instead of the lumbar spine’s angular extension velocity and reduced shear force. In the lumbar spine, rotation is restricted by the annulus anteriorly and the facet joints posteriorly [30]. Although the lumbar spine’s rotation ability is limited, this ability might also be affected by the flexibility of lumbar flexion–extension. Burnett et al. found that the range of the lumbar spine’s axial rotation decreased in end-range flexion and extension postures compared to the neutral spine posture [31]. This suggests that LS golfers may achieve the end of lumbar flexion or extension earlier than HS golfers due to the difference in the flexibility of the lumbar spine. Once the lumbar spine is at the end-range flexion position, the ability to rotate may be restricted. In addition, the axial loading of the spine in end-range flexion sagittal postures may result in a greater risk of injury if the soft tissues are loaded beyond their tolerance level, as this is the position where passive structures appear to be at their maximal stiffness [30]. Rotating beyond the point of soft-tissue tolerance may contribute to LBP by increasing the shear force acting on the intervertebral disc [32,33].

For accurate contact with the ball, increased trunk flexion is required to return the clubface to the initial setup position during the downswing at ball impact [34]. However, the lumbar spine will move toward extension before impact. This phenomenon may be explained with the proximal-to-distal sequencing theory [35], which proposes that for best energy transfer and maximum club head speed in the downswing, all segments should accelerate and then decelerate before impact with the ball. This kinematic sequence has been analyzed in golfers with different skill levels. Compared to amateur golfers, professional golfers show a slowing of the pelvis before impact, suggesting that pelvic deceleration before impact is a desirable trait for fast swings [36]. Golfers indeed tend to lock their pelvis before impact to generate a faster club head speed.

Grimshaw and Burden proposed that golfers might maintain a more stationary spine movement during the downswing after 3 months of core stability as well as spinal and hip mobility training [16]. By improving the stability of the core and the mobility of the spine and hip, the activity level of the erector spinae may be less in the HS group as these muscles are no longer required to generate a powerful eccentric contraction to decelerate the rapid motion of the trunk observed in the LS group [16]. This reduction in the muscle’s activation level may reduce lumbar vertebrae joint loads during the golf swing [9,21]. Lindsay and Horton found that while there was no significant difference in the peak lumbar extension angular velocity between healthy golfers and golfers with LBP, the magnitude of the peak lumbar angular velocity was slightly larger in golfers with LBP [37], which is consistent with the results obtained in this study for the HS and LS groups. Compared with the LS group, golfers in the HS group do not therefore need to have a relatively high opposite velocity to lock or stop their pelvis to generate a powerful swing because they have better mobility in their hip and spine joints and better core neuromuscular control. Furthermore, for the same amount of downswing time in both groups, the rapid velocity applied to the lumbar spine may cause a higher joint force in the LS group, which in turn may exacerbate symptoms of LBP.

### 4.3. Impact of the OHS on Lumbar Spine Kinematics in the Frontal Plane

After the top of the backswing, the spine should continuously bend toward the right side to hit the ball to the leading (left) side. The increased lateral bending on the lumbar segment’s trailing (right) side at impact may lead to spinal injuries [38]. Our results did not show a difference in the lumbar right-side bending angle at impact between the HS and LS groups. By comparison, the left hip’s internal rotation flexibility was found to be more suitable. Kim et al. found that the lumbar side bending angle at impact is different between golfers with and without limited hip internal rotation [38]. Golfers without this hip limitation demonstrated a smaller side bending angle at impact, and the left-hip internal rotation angle of golfers without this limitation was also greater than that of golfers with limitations at impact [38].

The lumbar side bending angular displacement during the golf downswing is purported to be associated with LBP. However, several studies have found no significant difference in the lumbar side bending angular displacement between golfers with and without LBP, or between those with and without limitations in left-hip internal rotation [37,38,39]. Cole and Grimshaw also found no significant difference in the lumbar right-side bending angle at impact among golfers with and without LBP [39]. This demonstrates that the right-side bending angle at impact alone may not be sufficient to fully characterize LBP risk.

Although the lumbar side bending angular displacement during the downswing does not appear to be a sensitive measure for distinguishing golfers with LBP from asymptomatic players, Grimshaw and Burden found that, after 3 months of coaching focused on improving the swing technique, a reduction in the amount of side bending during the downswing helped reduce LBP symptoms that may arise due to chronic overuse [16]. Furthermore, our results indicate that hip joint and spine joint restrictions have no effect on the lumbar side bending angular displacement during the downswing, which is consistent with the finding by Kim et al. [38]. This suggests that the side bending of the lumbar spine during the downswing is insufficient in revealing the risk of LBP, whereas using a physical ability test like the OHS shows that lateral bending during the golf downswing may be inappropriate.

Similar to our results, Lindsay and Horton found no differences in the right-side bending velocity of golfers with and without LBP [15]. This may be because of the club difference, where the 7-iron, for example, requires a more vertical swing plane, which may produce more lateral motion on the downswing than the driver. Furthermore, Bae et al. found that the lumbar flexion–extension joint power was significantly larger than that of the lumbar lateral bend during the downswing phase [40]. This indicates that the generation of lumbar rotation action during the downswing might be emphasized more in the sagittal plane than in the frontal plane.

### 4.4. Impact of the OHS on Lumbar Spine Loads

The golf swing is regarded as a three-dimensional movement, with restriction in the spine and hip mobility that may affect the lumbar joint loads and precipitate LBP. Significant differences were observed in this study between the HS and LS groups in L4-L5 and L5-S1 peak shear forces; in particular, our results indicate that golfers who had better performance in the OHS test also had lower loads applied to the lumbar spine during the downswing. These individuals may therefore have less risk of a golf-related lower back injury.

To the best of our knowledge, this is the first study to investigate the relationship between the overhead squat ability and lumbar spine loads during the golf downswing using computer simulation. The peak shear force applied to the L5-S1 joint in the LS group (525.19 ± 86.69, N) is similar to the result reported by Hosea et al. (596 ± 514, N) [21]. Furthermore, shear loads of similar magnitude (i.e., 570 ± 190 N) were found to result in pars interarticularis fractures with cyclic loading in cadaver specimens [41,42]. Sugaya et al. found that the golf swing’s asymmetrical pattern may cause the degeneration of the right lumbar spine around L4-L5 in right-handed golfers [43]. Moore and Dalley further suggested that most spinal disc herniations occur in the lumbar spine, with 95 percent at L4-L5 or L5-S1 [44]. The current study also revealed that in golfers with limitations in the overhead squat, a higher shear force may be generated in the L5-S1 and L4-L5 joints. Considering the fact that these two joints have the highest risk of injury, applying the OHS as a test tool may provide some useful information for coaches to prevent golf-related low back pain.

Lim et al. proposed that the L4-L5 and L5-S1 discs are subjected to large compressive loads due to the weight of the trunk and the muscle activity generated during the golf swing [22]. Hosea and his colleagues performed the first two studies that assessed joint loads during the golf swing. They found that compression loads were up to eight times the body weight, or about 6100 ± 2413 N in amateurs and 7584 ± 2422 N in professionals during the golf swing [9,21]. The compressive forces calculated in the present study, ranging from 3018.92 ± 233.43 N to 3921.69 ± 450.25 N, are lower than Hosea’s results but similar to the findings reported by Lim et al. [22]. Lim and colleagues proposed that the compressive load consistently increased after the top of the backswing and reached its maximum (4400 N or 6.1 BW) near ball impact [22]. Differences in these results may be due to differences in the methodology used to calculate the joint loads in these studies. This is especially the case in studies by Hosea et al. [21], where the measured ground reaction forces were not used in the calculation of the lumbar loads [22]. However, none of the above studies reported the compressive loads in each lumbar disc joint. Since the range of motion and stiffness value for each lumbar joint is different, it seems necessary to calculate the joint loads in each lumbar disc joint using computerized simulation methodology. As noted in the current study, the compressive load gradually decreased from S1 to L1 during the downswing phase, which agrees with a simulation study by Bae et al. [40]. This also indicates that the inferior region of the lumbar spine may be at a higher risk of injury than the superior region.

## 5. Conclusions

We investigated the differences in lumbar spine kinematics and joint loads during the golf downswing between golfers with different overhead squat abilities. Golfers with better performance in the overhead squat test demonstrated significantly greater angular extension displacement in the sagittal plane, lower lumbar extension angular velocity, and lower L4-S1 joint shear force than golfers with lower performance levels in the overhead squat test. Due to the requirements of performing the overhead squat, better performance in this test also reflects an advantage in hip and spine flexibility and core stability, which are associated with swing mechanics and the risk of LBP. The study’s findings, therefore, suggest that the overhead squat test can be a useful index in assessing the lumbar kinematics and joint loading patterns during the downswing, and the test provides a training guide reference to reduce the risk of a golf-related lower back injury. Applying a self-massage on the lateral torso, plantar fascia, and lateral thigh for 90 s or longer has been proven to acutely improve overhead deep squat scores [45]. Therefore, we suggest that executing a self-massage before or after a competition may reduce the spine load and the risk of LBP during the golf swing.

There are several limitations to the current study. First, different club types may affect lumbar swing kinematics [37,46] and hence the loads applied to the lumbar spine during the downswing phase. Further study is needed to investigate the relationship between the overhead squat ability and lumbar loads during the downswing when an iron club is used. Second, the highest L4-L5 shear loads were found after ball impact [22]. However, joint loads are also applied to the lumbar spine during the follow-through, and the end of the follow-through is also considered a critical element related to LBP in the golf swing [47]. Future work should examine lumbar kinematic and kinetic variables in the follow-through phase of the golf swing in order to extend the impact of the overhead squat ability on the lumbar joint loading.

Third, the current study only examined lumbar kinematics and joint loads in the sagittal and frontal planes. The axial rotation of the lumbar spine is associated with ball speed, although an over-rotated lumbar spine might result in excess loads that relate to LBP [21,48]. Rapid spinal rotation during the golf swing, combined with physical limitations, may play a role in golf-specific injuries [49]. Hence, additional research may prove beneficial for LBP prevention by investigating the difference in lumbar rotation kinematics between golfers with different overhead squat abilities. Finally, participants in the current study were healthy golfers free from any injuries that may have prevented them from performing the golf swing. To better understand the chronic impact of overhead squat ability on the risk of LBP during the golf swing, the next logical step is to investigate the difference in lower back injury rates between HS and LS golfers over time. In this way, the effect of the overhead squat test on LBP prevention can be studied.

## Figures and Tables

**Figure 1 sensors-24-01252-f001:**
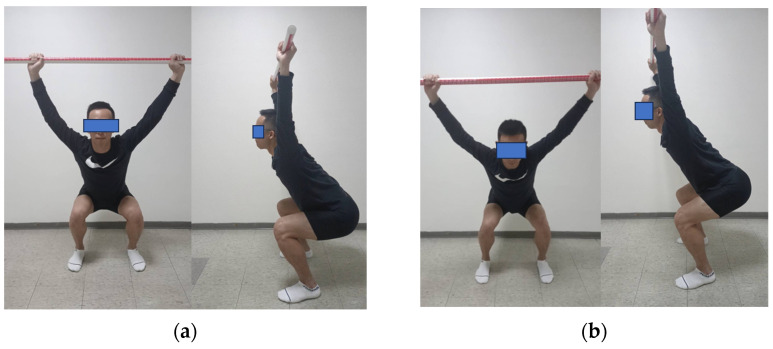
The squat movement examples of high-scoring group (**a**) and low-scoring group (**b**).

**Figure 2 sensors-24-01252-f002:**
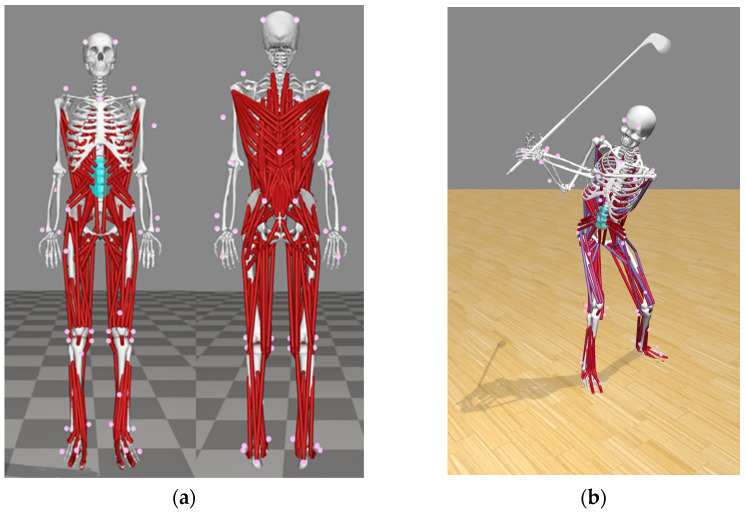
The full-body lumbar spine (FBLS) model (**a**) and the OpenSim simulation of golf swing (**b**).

**Figure 3 sensors-24-01252-f003:**
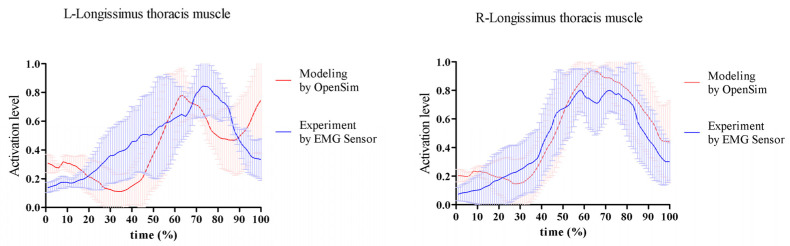
The model validation on activation level by measured normalized EMG RMS on both sides of the lumbar area.

**Table 1 sensors-24-01252-t001:** Scoring criteria used for the overhead squat (OHS).

Tests	3 Points	2 Points	1 Point	0 Points
Overhead Squat	Upper torso is parallel with tibia or toward vertical.	Meet criteria of 3 points with 2 × 6 board under heels.	Tibia and upper torso are not parallel.	If pain is associated with any portion of this test.
	Femur is below horizontal. Knees are aligned over feet. Dowel is aligned over feet.	Knees are not aligned over feet.	Femur is not below horizontal. Knees are not aligned over feet. Lumbar flexion is noted.	

**Table 2 sensors-24-01252-t002:** Subject characteristics.

	Mean (SD)		df	t	*p*
	LS-G (N = 11)	HS-G (N = 10)			
Downswing time (s)	0.29 (0.06)	0.30 (0.03)	19	−0.78	0.452
Height (m)	1.68 (0.10)	1.61 (0.05)	19	2.00	0.061
Weight (kg)	75.00 (14.37)	68.06 (13.67)	19	1.13	0.268
Best scores	68.45 (3.14)	69.90 (4.07)	19	−0.92	0.367
Ball speed (mph)	142.44 (16.95)	126.29 (11.52)	19	2.53	0.022 *

Note: * *p* < 0.05.

**Table 3 sensors-24-01252-t003:** Lumbar kinematics in the sagittal plane during golf downswing.

	Mean (SD)		df	t	*p*
	LS-G (N = 11)	HS-G (N = 10)			
Lumbar flexion angle at impact (°)	−21.37 (6.39)	−26.20 (5.90)	19	1.792	0.089
Lumbar angular extension displacement (°)	17.72 (5.94)	24.36 (7.11)	19	−2.33	0.031 *
Lumbar peak extension angular velocity (°/s)	119.52 (59.23)	40.51 (26.79)	19	3.87	0.000 ***
Lumbar peak flexion angular velocity (°/s)	−269.34 (181.03)	−288.95 (162.68)	19	0.26	0.798

Note: * *p* < 0.05, *** *p* < 0.001. Positive values indicate extension and negative values indicate flexion.

**Table 4 sensors-24-01252-t004:** Lumbar kinematics in the frontal plane during golf downswing.

	Mean (SD)		df	t	*p*
	LS-G (N = 11)	HS-G (N = 10)			
Lumbar right-side bending angle at impact (°) ^1^	21.07 (0.48)	20.93 (0.70)	19	0.55	0.591
Lumbar angular bending displacement (°) ^1^	34.89 (5.95)	31.21 (7.28)	19	−1.27	0.218
Lumbar peak right-side bending angular velocity (°/s) ^1^	329.35 (43.79)	287.42 (62.27)	19	1.80	0.088

Note: ^1^ Positive values indicate right-side bending.

**Table 5 sensors-24-01252-t005:** Lumbar joint peak shear force (N) during golf downswing.

	Mean (SD)		df	t	*p*
	LS-G (N = 11)	HS-G (N = 10)			
L1-L2	712.67 (74.46)	737.97 (133.89)	19	−0.54	0.594
L2-L3	487.45 (70.43)	530.31 (104.34)	19	−1.11	0.280
L3-L4	327.81 (75.54)	337.94 (56.10)	19	−0.35	0.733
L4-L5	387.19 (89.16)	299.54 (37.30)	19	2.88	0.010 **
L5-S1	525.19 (86.69)	407.90 (59.06)	19	3.59	0.002 **

Note: ** *p* < 0.01.

**Table 6 sensors-24-01252-t006:** Lumbar joint peak compressive force (N) during golf downswing.

	Mean (SD)		df	t	*p*
	LS-G (N = 11)	HS-G (N = 10)			
L1-L2	3018.92 (233.43)	3059.77 (356.55)	19	−0.31	0.757
L2-L3	3412.79 (289.78)	3515.89 (424.28)	19	−0.66	0.520
L3-L4	3668.80 (326.21)	3797.82 (443.92)	19	−0.76	0.454
L4-L5	3770.94 (338.66)	3921.69 (450.25)	19	−0.87	0.394
L5-S1	3759.18 (328.70)	3918.67 (447.88)	19	−0.94	0.361

## Data Availability

All data are contained within the manuscript.

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
