# Peer review of "Does Overhead Squat Performance Affect the Swing Kinematics and Lumbar Spine Loads during the Golf Downswing?"

_sensors, 2024, doi:10.3390/s24041252_

Round 1

Reviewer 1 Report

Comments and Suggestions for Authors

Dear authors,

First of all, thank you for the opportunity to review this article.  The present article This study investigates the differences in lumbar kinematics and joint loads during the golf downswing between golfers with different overhead squat abilities.

In general, the article is very clearly elaborated. It uses a computer modelling and simulation approach to calculate lumbar joint forces during the golf downswing, the inclusion of both kinematic and kinetic data to provide a comprehensive understanding of lumbar spine loads, and the investigation of the relationship between overhead squat performance and golf swing mechanics, proving that this is a well conducted study; the aim and object of the research is understandable.

The results could be better presented: I found it difficult to read figure 3. Also, I would like to see the effect size calculated in your results.

In the discussion, check misspellings in line 261 and figure 3: lumber – lumbar. Also, this sentence is not very clear, please rephrase.

Lines 266-271 seem a bit confusing to me. You state that golfers with poor OHS may have inefficient golf swing, but I think you do not have data to prove it. In fact, their ball speed is higher, and no other parameters are different.

Then, the last sentence of this paragraph is not clear to me: Please explain how the OHS can be an indicator of efficiency of the golf game. Which metrics did you use to measure golf efficiency and establish this connection?

The conclusions clearly state the limitations of this study and occupy a big part of the section. I believe you should highlight a bit more the practical implications of your paper, how can this information be used to improve training’

Regarding the references, I think you are not following the Journal’s guidelines. Please confirm.

In my humble opinion, this is a suitable work for the journal and especially for the special issue Human Movement Monitoring Using Wearable Sensor Technology.

Finally, I would like to congratulate the authors for the work. I hope my comments and remarks helped!

Comments on the Quality of English Language

Language is fine, Just check some minor errors.

Reviewer 2 Report

Comments and Suggestions for Authors

Reviewer comments

The manuscript, entitled "Relationship between overhead squat motion and spinal Lum-2bar load during Golf Downswing" aims to explore how performance on the elevated squat test affects low back pain associated with golf swing mechanics. In the opinion of the reviewers, the most outstanding highlight of the current study is the proposal of the elevated squat test as an indicator of the waist movement and joint load pattern. If the test can accurately reflect the biomechanical characteristics of a golfer's swing, it could provide new guidance for future training and prevention.

Specific suggestions are as follows:

introduction:

 The reviewers believe that the background, questions, purpose and methods of the research need to be more clearly expressed to ensure that readers can understand the scientific and innovative nature of the research. Moreover, the content lacks literature support, and the relevant literature on the relationship between golf swing and low back injury needs to be supplemented. This paragraph also does not explain how the elevated squat test actually helps the golf swing. The role of the elevated squat test needs to be explained in more detail as to why the test is helpful in assessing the golf swing and how it reflects the biomechanical characteristics of the athlete.

Lines 29-31: The start section introduces golf as a sport that is growing in popularity around the world and its positive effects on health. However, this section could more specifically introduce the research question of the problem of low back injuries in golf, and how this affects athletes.

Lines 37-38: The research question needs to be stated more clearly. Why study the association between golf swing and low back injuries? Questions about how the elevated squat test relates to the golf swing need to be articulated more clearly.

Lines 100-102: Some of the methods used to measure lumbar joint load are described, but a more detailed explanation of the options and advantages of these methods may help readers understand the science of the study design.

Methods:

Lines 116-117: What is the level of the athletes and what are the criteria for selecting the 21 athletes? Why are these 21 athletes selected for the experiment? After all, the physical functions and conditions of athletes of different levels and ages vary greatly, and the reviewers suggest that the authors describe them clearly.

Line 131-132: Whether the accuracy of the experimental results will be affected by the subjects wearing their own sneakers? The difference in the material of the shoes may lead to the deviation of the experimental results. If it does not affect the accuracy of the experimental results, the author should specifically explain.

Lines 189-191: The authors used a low-pass, sixth-order Butterworth digital filter and identified a cutoff frequency of 14 Hz, could a more detailed explanation be provided as to why this particular filter type and parameter was chosen? Or have other cutoff frequency tests or sensitivity analyses been performed?

Lines 203-205: In calculating the compressive and shear forces of the intervertebral joint, the author mentions the direction of the forces. Can more detailed information be provided to ensure that the coordinate system and orientation definitions used in these calculations are consistent and comply with the laws of anatomy and biomechanics?

Lines 245-246: It is mentioned that HS has significantly lower Shear forces at L4-L5 and L5-S1. Do the authors discuss the possible biomechanical effects of these lower Shear forces, and are they consistent with the context of the study question?

Lines 249-250: The Compressive forces of L1-L2, L2-L3, L3-L4, L4-L5, and L5-S1 are mentioned in the results. Can more explanation and background information be provided on these Compressive forces results?

Lines 254-255: The results indicate that there is no significant difference in golf scores between the two groups, even if there is no significant difference. The author can provide a more in-depth explanation for why golfers with poor OHS ability have an advantage in speed but no significant difference in overall golf performance?

Lines 278-280: The authors mention changes in core stability and activity levels in the spine and hips, but do not seem to discuss how these changes affect load in the lumbar spine and potential LBP risk.

Lines 346-348: The authors mention lateral bending velocity as an alternative to lateral bending data, but do not discuss the feasibility and validity of such an alternative measurement, nor does it seem to take into account the role of lateral bending velocity in LBP risk

Lines 362-363: The conclusion mentions that OHS may provide coaches with information on how to prevent golf-related back pain. The reviewer suggests that more specific practical application suggestions can be provided. Or whether the potential risk of lumbar injury was explored.

Conclusion:

Lines 400-402: It is mentioned in the conclusion that players who perform better on the high leg lift squat test show better lumbar movement and lower joint load during the swing. It seems that the authors did not take the proper approach in the study design and analysis to confirm this cause-and-effect relationship, only based on correlation.

Round 2

Reviewer 1 Report

Comments and Suggestions for Authors

Dear Authors,

Thank you for yor responses.

I am satisfied with your answers and have nothing else to add.

Congratulations on the work!